# GER: GENERATION, EVALUATION AND REFLECTION ENHANCED LLM FOR KNOWLEDGE GRAPH QUESTION ANSWERING

## ABSTRACT

Knowledge Graph Question Answering (KGQA) involves answering natural language questions based on information provided by knowledge graphs. Large language models (LLMs), utilizing their exceptional natural language understanding capabilities and factual knowledge from knowledge graphs, have made some progress in KGQA reasoning. However, existing methods overlook the amplification of hallucinations in large language models caused by irrelevant information within vast knowledge graphs. This oversight leads to answers containing seemingly correct but unrelated responses, decreasing reliability. In this paper, we propose *Generation-Evaluation-Reflection* (Ger), an LLM-enhanced reflective reasoning framework for KGQA. The Ger mechanism introduces evaluation and reflection steps during the reasoning process, enabling LLMs to better utilize factual information in knowledge graphs for assessing and correcting their answers. This process systematically reduces errors and hallucinations while improving the reasoning accuracy of LLMs. Extensive experiments on multiple KGQA benchmark datasets demonstrate that Ger enhances reasoning performance, providing more reliable and interpretable results, and achieves new state-of-the-art levels.

## 1 INTRODUCTION

Knowledge Graph Question Answering (KGQA) is a task to answer questions expressed in natural language through reasoning over the entities and relationships in a knowledge graph (KG). Knowledge graphs store vast amounts of information in triples (e.g., Freebase, Wikidata (Vrandečić & Krötzsch, 2014)), providing factual information as a basis, which plays a crucial role in applications requiring reliability. Due to this importance, KGQA is widely researched and applied in customer service, information retrieval systems, and more, such as Google, Apple Siri, and Microsoft Cortana (Lan et al., 2022). KGQA faces two main challenges: comprehensively understanding the user intent embedded in the natural language questions and having the capability to perform correct multi-hop reasoning over large-scale knowledge graphs without being misled to derive the correct answer from thousands of candidates (Lan et al., 2022).

Classical works (Sun et al., 2018; Shi et al., 2021; He et al., 2021; Jiang et al., 2022), such as UniKGQA (Jiang et al., 2022), typically use GNN-based methods to prune and retrieve a question-specific graph that provides information related to the question and predict the correct answer from a large pool of candidates based on the question. While GNN-based methods effectively adapt to the network structure of KGs, they often lack natural language understanding capabilities, leading to potential misinterpretation of the question intent. Recently, large language models (LLMs) (Brown et al., 2020; Chowdhery et al., 2023; Touvron et al., 2023; Achiam et al., 2023) have demonstrated powerful capabilities in tasks related to natural language understanding.

By combining LLMs with KGs and utilizing the instruction-following capabilities of LLMs, some existing works can achieve more relevant logical reasoning related to the questions and have made significant progress. However, LLMs are often criticized for having a "hallucination" problem (Bang et al., 2023; Pan et al., 2024), where they output seemingly correct but factually incorrect responses. Research has shown that this phenomenon could be influenced by the input content of LLMs: the more irrelevant and disordered the input, the more severe the hallucination. Existing methods that

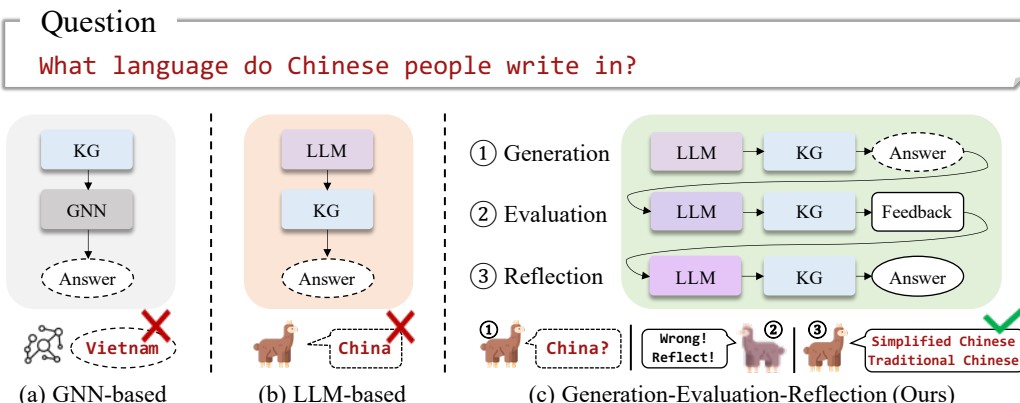

Figure 1: Current categorization of KGQA methods. **(a) GNN-based** approaches operate directly on the knowledge graph, using embeddings to predict answers. **(b) LLM-based** methods leverage knowledge graph triplets to generate answers. **(c) Ours** methods extend beyond simple LLM reasoning by incorporating evaluation and reflection steps, which correct erroneous answers and mitigate hallucinations.

integrate LLMs with KGs often overlook the impact of the vast information in knowledge graphs fed to LLMs, resulting in many hallucinatory answers being generated alongside the correct answers, posing new challenges to applications that rely on high reliability.

For instance, RoG (Luo et al., 2024) leverages LLMs to generate inference paths and answers to final questions, yet this method often produces outputs mixed with numerous hallucinations. Similarly, the EtD (Liu et al., 2024) method utilizes LLMs to generate answers based on pruned knowledge graphs processed by GNNs, leading to responses plagued by hallucination issues. The inherent complexity of entities and relationships within knowledge graphs exacerbates the hallucination problem in KGQA, highlighting the necessity for a systematic solution.

This paper introduces a framework named ***Generation-Evaluation-Reflection*** (`Ger`). This framework is composed of three modules: generation, evaluation, and reflection. It systematically reduces hallucinated answers and reflects on errors to explore correct answers. The generation module utilizes large language models to explore the knowledge graph and produce preliminary answers, which may contain hallucinations and partial errors. The evaluation module then assesses the completeness and correctness of the answers through a dual-granularity evaluation strategy. Based on the evaluation results, the reflection module guides the LLM to filter potential hallucinations and explore missing answers, yielding the final answer. These three modules work collaboratively to enhance the reliability and precision of the LLM in knowledge graph question answering tasks. Our contributions are summarized as follows:

- We introduce a novel evaluation and reflection framework (`Ger`) for KGQA that combines knowledge graphs and LLMs to systematically reduce the numerous hallucinations, improving overall answer accuracy.

- We design a dual-granularity evaluation mechanism that not only removes individual hallucinated answers but also assesses answer completeness in the context of the questions, thereby enhancing the reliability and integrity of LLMs in knowledge graph question answering.

- Extensive experiments demonstrate that our method significantly improves the hit rate of correct answers, surpassing existing state-of-the-art methods and achieving efficient and interpretable reasoning over the knowledge graph.

## 2 RELATED WORK

**KG-enhanced LLM.** KG-enhanced LLM is a method that combines knowledge graphs (KG) with large language models (LLM) to improve reasoning capabilities. Early research attempted to enhance model performance by embedding structured knowledge from KGs into the pre-training or fine-tuning process of LLMs (Xie et al., 2023). However, this approach often led to reasoning errors

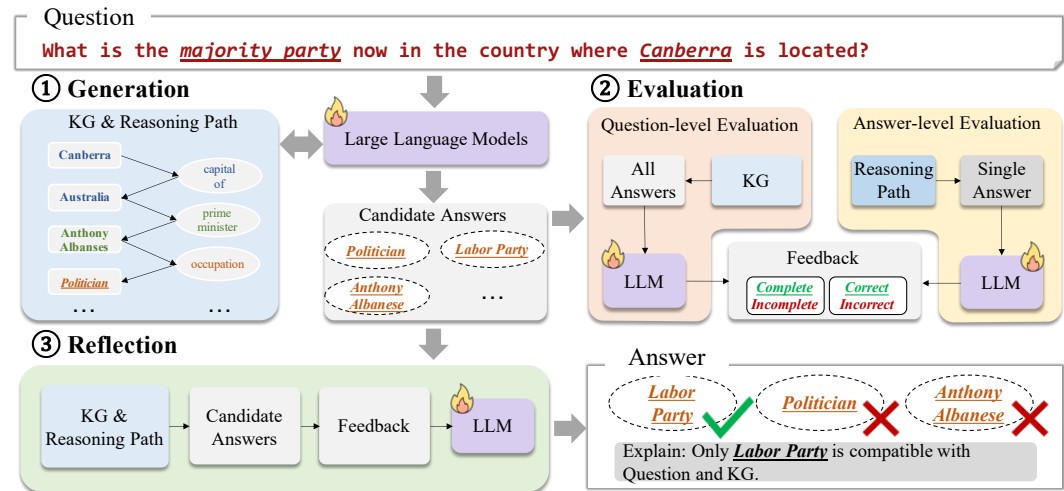

Figure 2: The overall framework of Generation, Evaluation, and Reflection (Ger). **(1) Generation:** Given a question, we employ the large language model to generate candidate answers based on the knowledge graph. **(2) Evaluation:** Then, we evaluate the completeness and correctness of the overall answer as well as individual answers based on these candidate answers and reasoning paths. **(3) Reflection:** Finally, based on the evaluation and the question, we guide the model to regenerate the answer.

and weakened the interpretability of KGs and the flexibility of knowledge updates. Researchers proposed using LLMs as agents for dynamic reasoning with external KGs to overcome these issues. For example, the ToG (Sun et al., 2023) model helps reasoning by having LLMs score relation-entity pairs in KGs. Additionally, prompt engineering further improved the reasoning abilities of LLMs by designing appropriate prompts that enable LLMs to generate plans and reason step-by-step (He et al., 2022). However, in the above methods, LLMs still face the issue of "hallucination." Especially the incorporation of KGs introduces a large amount of irrelevant information contained in the KGs, along with the lack of clearly defined relationships and entities, exacerbates the hallucination problem in LLMs.

**Knowledge Graph Question Answering (KGQA).** KGQA (Knowledge Graph Question Answering) is evolving from traditional GNN (Graph Neural Network) embedding methods to more sophisticated reasoning models. Early approaches, such as (Miller et al., 2016; He et al., 2021; Yasunaga et al., 2021), used GNN embeddings for entities and relations to score and rank answers, but struggled in handling complex semantics and higher-order connections. Retrieval-augmented methods assist LLM reasoning by retrieving triples, although they often introduce irrelevant information (Jiang et al., 2022). Agent-based approaches enable iterative interaction between knowledge graphs and LLMs, but are limited by LLMs' restricted graph comprehension and high computational costs (Yao et al., 2022). Some work combines the representational capabilities of GNNs with the language abilities of LLMs to unify retrieval and reasoning Liu et al. (2024). However, current methods overlook the impact of a large amount of irrelevant information in KGs on the hallucination problem in LLMs, such as the thousands of relations connected to a single entity and entity and relation definitions lacking self-explanatory capabilities, for example, (entity m.01wxmyl). Consequently, the risk of amplified hallucination phenomena in LLMs is present, leading to decreased reliability of LLMs in KGQA.

## 3 APPROACH

In this section, we introduce our method: *Generation-Evaluation-Reflection* (Ger), which consists of three components: 1) a *Generation* module that generates preliminary answers to questions based on knowledge graphs and LLMs; 2) an *Evaluation* module that assesses the correctness of answers based on reasoning paths in the KG; and 3) a *Reflection* module that guides the LLM to eliminate

hallucinations and generate new responses based on the evaluation results. The overall framework of `Ger` is illustrated in Figure 2.

## 3.1 GENERATION-EVALUATION-REFLECTION

Recently, numerous techniques have been developed to address Knowledge Graph Question Answering (KGQA) by integrating knowledge graphs with LLMs. However, the large amount of irrelevant information often present in KGs can exacerbate the issue of hallucinations in LLMs, resulting in inaccurate plans and incorrect answers (Ji et al., 2023).

To tackle this challenge, we propose a novel *Generation-Evaluation-Reflection* (`Ger`) framework. This framework leverages knowledge graphs to validate and optimize the responses generated by LLMs, systematically minimizing the impact of irrelevant information in KGs on LLM hallucinations.

By treating relation paths as plans, we ensure that these plans are grounded in knowledge graphs, allowing the LLMs to perform accurate and interpretable reasoning. In essence, we frame `Ger` as an optimization problem, aiming to maximize the likelihood of delivering accurate answers from a knowledge graph $\mathcal{G}$ in response to a question $q$. This is achieved by generating an initial answer $a'$, evaluating it to obtain an evaluation result $e'$, and reflecting on these insights to produce a final answer $a$

$$P_\theta(a|q, \mathcal{G}) = \sum_{a'} \sum_{e'} P_{\text{gen}}(a'|q, \mathcal{G}; \theta) P_{\text{eval}}(e'|q, \mathcal{G}, a'; \theta) P_{\text{ref}}(a|q, \mathcal{G}, a', e'; \theta), \quad (1)$$

where $\theta$ denotes the parameters of LLMs, $a'$ denotes the initial answer generated by the model, and $e'$ denotes the evaluation result of the initial answer. The first term $P_{\text{gen}}(a'|q, \mathcal{G}; \theta)$ is the probability of generating an initial answer $a'$ given the question $q$ and the knowledge graph $\mathcal{G}$, which is realized by the *generation* module. The middle term $P_{\text{eval}}(e'|q, \mathcal{G}, a'; \theta)$ is the probability of obtaining an evaluation result $e'$, which is computed by the *evaluation* module. The last term $P_{\text{ref}}(a|q, \mathcal{G}, a', e'; \theta)$ is the probability of generating the final answer $a$, which is computed by the *reflection* module.

## 3.2 GENERATION MODULE

The generation module is designed to produce potential candidate answers to questions. To leverage the instruction-following capability of LLMs, we follow Rog (Luo et al., 2024) by utilizing prompts to guide the large model in generating reasoning paths on the knowledge graph based on the given question.

> Please generate a valid relation path that can be helpful for answering the following question:
> `<Question>`

where `<Question>` indicates the question $q$. The question, together with the instruction template, is fed into LLMs to generate the relation paths, which are structurally formatted as a sentence:

$$z = \texttt{<PATH>}\ r_1\ \texttt{<SEP>}\ r_2\ \texttt{<SEP>} \ldots \texttt{<SEP>}\ r_l\ \texttt{</PATH>}$$

where `<PATH>`, `<SEP>`, `</PATH>` are special tokens indicating the start, separator, and end of the relation path, respectively. The retrieval process can be conducted by finding paths in $\mathcal{G}$ that start from the question entities and follow the relation paths $z$ to get reasoning paths $\mathcal{W}_z$ (including both entity and relationship). Similarly, we design a reasoning instruction prompt to guide LLMs to generate initial answers to question $q$ based on the retrieved reasoning paths $\mathcal{W}_z$ to get candidates' answers.

## 3.3 EVALUATION MODULE

The evaluation module primarily assesses the reliability and quality of candidate answers. To prevent the omission of answers and to identify hallucinated responses as much as possible, we designed two granularity levels of evaluation methods: Question-level Evaluation and Answer-level Evaluation.

**Question-level Evaluation** assesses the overall answer to the question. By inputting the question, reasoning path, and all candidate answers, we construct evaluation prompts to ask the LLM to provide feedback on whether the overall answer is correct and try to detect omissions.

> Please assess whether the candidate answers, derived from the reasoning paths, are entirely correct, and briefly explain the reasoning: `<Question>` `<Knowledge Graph>` `<Candidate Answers>`

where `<Question>` indicates the question $q$, `<Knowledge Graph>` are the related graph that helps reveal logical connections, and `<Candidate Answers>` are the outputs generated from these navigational insights. The knowledge graph and the instruction template are fed into LLMs to evaluate the whole answer, detect severe wrong answers, and determine if any answer is absent.

To formalize the Question-level Evaluation as an optimization problem as

$$\arg\max_{\theta} \frac{1}{n}\sum_{i=1}^{n} \log P_\theta(a_i|q,\mathcal{G}) = \frac{1}{n}\sum_{i=1}^{n}\log\prod_{j=1}^{|a_i|} P_\theta(w_j|w_{<j},q,\mathcal{G}), \quad (2)$$

Where $P_\theta(a_i|q,\mathcal{G})$ denotes the probability of candidate answer $a_i$ being correct given the question $q$ and the knowledge graph $\mathcal{G}$, $P_\theta(w_j|w_{<j},q,\mathcal{G})$ represents the probability of each word or element $w_j$ in answer $a_i$, conditioned on the preceding elements $w_{<j}$, the question, and the knowledge graph. $n$ is the total number of candidate answers.

**Answer-level Evaluation** assesses whether the reasoning process for individual answers is reasonable and filters out illogical, hallucinated answers. By constructing evaluation prompts, we similarly ask the LLM to indicate whether a single answer is correct. LLM does not need to provide detailed analysis when responding to ensure efficiency.

> Please assess whether the answer and reasoning path are logically correct for the question: `<Question>` `<Reasoning Path>` `<Answer>`

Similarly, to formulate the Answer-level Evaluation as

$$\arg\max_{\theta} \log P_\theta(a|q,r,\mathcal{G}) = \log\prod_{j=1}^{|a|} P_\theta(w_j|w_{<j},q,r,\mathcal{G}) \quad (3)$$

Where $P_\theta(a|q,r,\mathcal{G})$ indicates the probability of the individual candidate answer $a$ being correct given the question $q$, the reasoning path $r$, and the knowledge graph $\mathcal{G}$. This probability encapsulates whether the reasoning and answer are logical and accurate, $\prod_{j=1}^{|a|} P_\theta(w_j|w_{<j},q,r,\mathcal{G})$ is the product of probabilities for each element in the candidate answer sequence, ensuring that each part is logical and consistent with the reasoning path and question context.

The optimization problem, composed of two evaluation methods, focuses on ensuring both the completeness of the overall answer and the logic and accuracy of each response. Only answers that are both complete and supported by appropriate reasoning paths can be effectively applied in real-world applications that demand high reliability.

## 3.4 REFLECTION MODULE

The reflection module is designed to leverage evaluation results to eliminate hallucinations and identify missing correct answers. The inherent limitation of language models (LLMs) trained solely on QA data is their inadequate capacity to process feedback and adapt accordingly. To address this challenge, we propose a feedback-based training paradigm to enhance the LLM's capability to detect hallucinated answers and reassess and accurately locate the correct answers within the knowledge graph from its erroneous outputs.

> Based on the feedback from questions and answers, generate new answers.: `<Question>` `<Reasoning Path>` `<Answer>` `<Evaluation>`

To formally define the optimization problem for the feedback-based training paradigm using input answer feedback, question feedback, a question, and a reasoning path, we can construct an argmax formulation. The optimization problem can be formulated as

$$\arg\max_{\theta} \log P_\theta(a|q,\mathcal{G},a',e') = \log \prod_{j=1}^{|a|} P_\theta(w_j|w_{<j},q,\mathcal{G},a',e'), \tag{4}$$

where $P_\theta(a|q,\mathcal{G},a',e')$ represents the probability of generating the final answer $a$ given the question $q$, the knowledge graph $\mathcal{G}$, the initial answer $a'$, and the evaluation feedback $e'$. In this context, $w_j$ denotes each word or component in the final answer $a$, and $w_{<j}$ represents the sequence of preceding words or components. The product $\prod_{j=1}^{|a|} P_\theta(w_j|w_{<j},q,\mathcal{G},a',e')$ thus captures the probability of each element being correct and logically consistent with the inputs, thereby ensuring the final answer is refined and reliable according to the feedback received.

This novel paradigm shifts traditional QA training methodologies by focusing on dynamic learning through feedback, facilitating the LLM's development into a more robust and accurate answer generator. Through the reflection module, our approach ensures continuous improvement in LLM performance, fostering a more reliable alignment between questions, contextual understanding, and the knowledge graph.

## 4 EXPERIMENT

### 4.1 EXPERIMENT SETTINGS

**Datasets.** We assess the performance of `Ger` using two benchmark KGQA datasets: WebQuestionSP (WebQSP) (Yih et al., 2016) and Complex WebQuestions (CWQ) (Talmor & Berant, 2018). These datasets involve extracting subgraphs from Freebase (Bollacker et al., 2008) based on entities mentioned in up to 4-hop questions. The WebQSP dataset includes 2,826 questions for training and 1,628 for testing, while the CWQ dataset comprises 27,639 questions for training and 3,531 for testing.

**Evaluation Metrics.** Consistent with previous research, we employ Hits@1 and F1 as evaluation metrics. Hits@1 evaluates the percentage of questions for which the top-1 predicted answer is correct. Since a question may have multiple correct answers, F1 accounts for the coverage of all possible answers, balancing the precision and recall of the predicted answers. For large language models (LLMs), due to the autoregressive nature of answer generation, which may not preserve the order of predicted answers, we align with Luo et al. (2024) in calculating the Hits@1 metric.

**Implementation Details.** For the `Ger` evaluation, we use LLaMA2-Chat-7B (Touvron et al., 2023) as the foundational LLM and apply instruction fine-tuning over 3 epochs using the WebQSP and CWQ training datasets. Given that Rog (Luo et al., 2024) represents the current state-of-the-art in this domain, we reference the results from their study along with those of other baselines for comparative purposes.

### 4.2 KGQA PERFORMANCE COMPARISON

**Main Results.** This section presents a comparison between `Ger` and other baseline models in Knowledge Graph Question Answering (KGQA) tasks, with results summarized in Table Table 1. Our method consistently achieves superior performance across most metrics on both datasets. Specifically, compared to the baseline method Rog, our approach improves Hits@1 by $5.0\%$ and F1 by $3.5\%$ on the WebQSP dataset, establishing a new state-of-the-art. On the more challenging CWQ dataset, which includes more complex knowledge graph information, our method outperforms the state-of-the-art model with a $6.8\%$ improvement in Hits@1 and a $1.5\%$ increase in F1. These results underscore our method's proficiency in accurately answering questions within knowledge graphs rich in entities.

Table 1: Performance comparison with different baselines on the two KGQA datasets.

| Type | Methods | WebQSP | | CWQ | |
|---|---|---|---|---|---|
| | | Hits@1 | F1 | Hits@1 | F1 |
| GNNs | KV-Mem (Miller et al., 2016) | 46.7 | 34.5 | 18.4 | 15.7 |
| | EmbedKGQA (Saxena et al., 2020) | 66.6 | - | 45.9 | - |
| | NSM (He et al., 2021) | 68.7 | 62.8 | 47.6 | 42.4 |
| | TransferNet (Shi et al., 2021) | 71.4 | - | 48.6 | - |
| | KGT5 (Saxena et al., 2022) | 56.1 | - | 36.5 | - |
| | GraftNet (Sun et al., 2018) | 66.4 | 60.4 | 36.8 | 32.7 |
| | PullNet (Sun et al., 2019) | 68.1 | - | 45.9 | - |
| | SR+NSM (Zhang et al., 2022) | 68.9 | 64.1 | 50.2 | 47.1 |
| | SR+NSM+E2E (Zhang et al., 2022) | 69.5 | 64.1 | 49.3 | 46.3 |
| | SPARQL (Sun et al., 2020) | - | - | 31.6 | - |
| | QGG (Lan & Jiang, 2020) | 73.0 | 73.8 | 36.9 | 37.4 |
| LLMs | Flan-T5-xl (Chung et al., 2022) | 31.0 | - | 14.7 | - |
| | Alpaca-7B (Taori et al., 2023) | 51.8 | - | 27.4 | - |
| | LLaMA2-Chat-7B (Touvron et al., 2023) | 64.4 | - | 34.6 | - |
| | ChatGPT | 66.8 | - | 39.9 | - |
| | ChatGPT+CoT | 75.6 | - | 48.9 | - |
| LLMs+KGs | G-Retriever(He et al., 2024) | 70.1 | - | - | - |
| | KD-CoT (Wang et al., 2023) | 68.6 | 52.5 | 55.7 | - |
| | UniKGQA (Jiang et al., 2022) | 77.2 | 72.2 | 51.2 | 49.1 |
| | ToG+ChatGPT(Sun et al., 2023) | 76.2 | - | 58.9 | - |
| | EtD+ChatGPT(Liu et al., 2024) | 82.5 | - | 62.0 | - |
| | Rog (Luo et al., 2024) | 85.7 | 70.8 | 62.6 | 56.2 |
| | Ger (**Ours**) | **90.7** | **74.3** | **69.8** | **57.7** |

Other methods, particularly those solely relying on LLMs, display considerable performance variability. On the WebQSP dataset, there is nearly a 40% disparity in the Hits@1 metric between the best and worst-performing methods, with a similar 35% gap observed on CWQ. This suggests inherent differences in KGQA capabilities among LLMs, and given that non-fine-tuned LLMs often produce uncontrolled outputs, we follow Rog's approach by reporting only the Hits@1 metric.

Comparing methods that integrate LLMs with knowledge graphs (LLM+KGs) to those using only LLMs, we observe superior performance in the former. This highlights the efficacy of combining knowledge graphs and large language models for reasoning tasks. Additionally, our method shows notably enhanced performance within the LLM+KGs paradigm, demonstrating the effectiveness of the Evaluation and Reflection mechanisms in this framework. Specific cases can be referenced in Figure 3.

## 4.3 ABLATION

### 4.3.1 EFFECTIVENESS OF EVALUATION

We conducted ablation studies to analyze the effectiveness of the evaluation module in our approach (Ger). We compared two variants: 1) *w/o reflection*, where the evaluation module is removed, and no evaluation information is provided to the reflection module; 2) *w/ negative*, where all results are forced to be evaluated as negative, requiring the reflection module to reflect on each answer. The results are shown in Table 2. From the table, it is evident that the absence of effective evaluation information leads to a significant decline in performance. Specifically, there is a 4.56% decrease in performance on the WebQSP dataset when no evaluation information is provided to the reflection module compared to the Ger method, demonstrating the effectiveness of the evaluation and reflection mechanism. Additionally, in both WebQSP and CWQ datasets, forcing reflection results in more than 10% drop in the F1 metric, indicating that relying solely on the reflector does not lead to actual performance improvement.

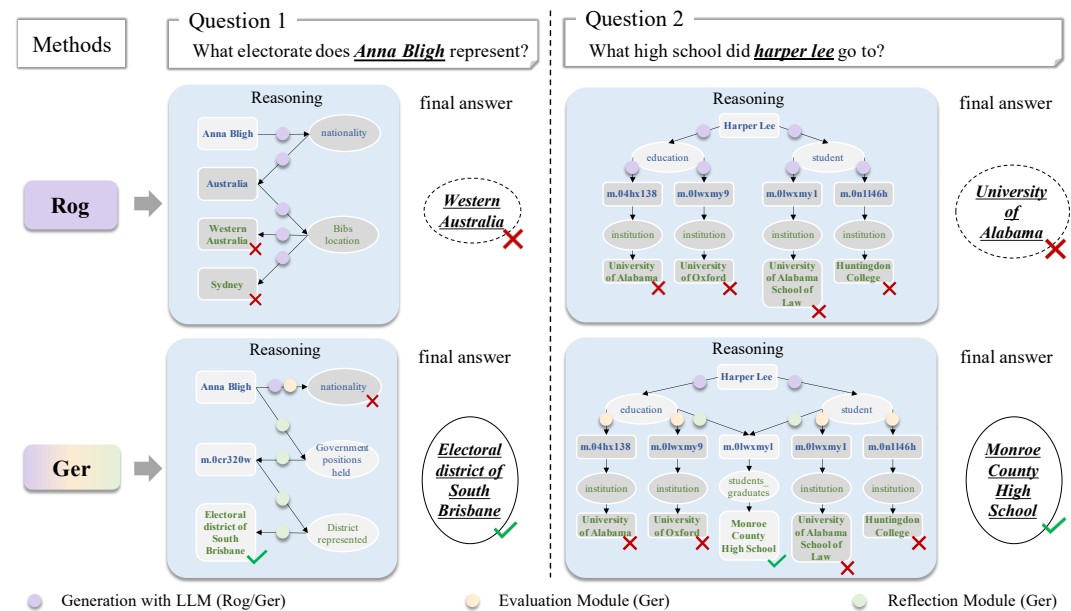

Figure 3: Two case studies that illustrate how Ger improves the LLM's reliability.

### 4.3.2 EFFECTIVENESS OF REFLECTION

In this section, we evaluate the effectiveness of integrating the reflection module with Ger. Specifically, we set up two variants: 1) *w/o reflection*, using the evaluation results from the evaluation module as the reflection results. 2) *w/ generation*, where we directly use the generation module to receive feedback from the evaluation module and output the results as predictions.

Table 2: Ablation studies of Ger.

| Method | WebQSP | | | CWQ | | |
|---|---|---|---|---|---|---|
| | Precision | Recall | F1 | Precision | Recall | F1 |
| Ger | **81.11** | 80.10 | **74.32** | **57.35** | 65.89 | **57.73** |
| Ger w/o evaluation | 73.58 | 76.42 | 70.25 | 56.88 | 57.87 | 55.40 |
| Ger w negative | 54.44 | **87.69** | 60.51 | 25.43 | **73.58** | 35.50 |
| Ger w/o reflection | 77.76 | 72.54 | 69.66 | 46.38 | 44.58 | 43.36 |
| Ger w/ generation | - | - | - | - | - | - |

The results are shown in Table 2. From the table, it is clear that the lack of an effective reflector leads to a significant drop in performance. On the CWQ dataset, relying solely on the evaluation module to determine the final answer results in decreases in both precision and recall, with the F1 metric dropping by 14.37%. 2) The results for the *w/ generation* part are empty because we found that the model used in the first stage for generating answers is not able to adapt to the paradigm of receiving evaluation information for further answer generation, leading to a collapse in output patterns and meaningless repetitive token outputs.

### 4.4 CASE STUDIES ON EVALUATION AND REFLECTION

Figure 3 presents two case studies from the WebQSP dataset, illustrating how Ger improves the accuracy of LLM-generated responses by correcting errors through evaluation and reflection. In the first case, the baseline Rog (Luo et al., 2024) method's LLM produces an incorrect reasoning path <Anna Bligh → nationality → Australia → Western Australia, Sydney>, leading to a wrong answer. Ger identifies the candidate answer Western Australia, Sydney as incorrect through evaluation and then uses a reflection mechanism to derive the correct reasoning path and answer, Electoral district of South Brisbane. In the second case, the Rog baseline misses the path <Harper Lee → education → m.0lwxmyl → students_graduates → Monroe County High School>. The evaluation module identifies that the correct answer was overlooked, and the reflection module captures the missing path, resulting in the accurate answer Monroe County High School.

## 5 CONCLUSION

This paper presents the Generation-Evaluation-Reflection (`Ger`) framework, specifically designed to improve large language models (LLMs) in knowledge graph question answering. By systematically integrating generation, evaluation, and reflection components, `Ger` enhances the logical consistency and factual accuracy of outputs, effectively minimizing hallucinations that are amplified by irrelevant facts from the knowledge graph in LLM reasoning processes.

Despite these advancements, `Ger` faces certain limitations. Its dependency on the completeness and accuracy of the underlying knowledge graph can limit its performance in scenarios where the graph data is sparse or erroneous. Moreover, including additional evaluation and reflection stages introduces a layer of computational complexity, potentially affecting scalability and response time for complex queries. Addressing these areas in future work could involve developing more efficient algorithms and exploring strategies to handle incomplete graph data more effectively.

Experimental evaluations demonstrate that `Ger` achieves higher accuracy and reliability in KGQA tasks than existing methodologies, providing a robust framework for leveraging LLMs in knowledge-driven applications. Future research directions include optimizing the framework's computational efficiency and extending its applicability to more diverse and challenging datasets.

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
