# OpenReview forum: "Ger: Generation, Evaluation and Reflection Enhanced LLM for Knowledge Graph Question Answering"
_ICLR.cc/2025/Conference — Submitted to ICLR 2025_

### Official Review · Reviewer_UZng · 2024-10-16

**Soundness:** 3
**Presentation:** 2
**Contribution:** 2
**Rating:** 3
**Confidence:** 5

**Summary:**

In the path-based KGQA solution of RoG, evaluation and reflection processes utilizing the reasoning abilities of large language models were introduced. A new Ger framework, which incorporates these evaluation and reflection processes, was proposed to overcome the limitations of RoG and achieve state-of-the-art performance on the WebQSP and CWQ datasets.

**Strengths:**

1. It overcame the limitations of the existing RoG and achieved a high level of KGQA performance.
2. A new dual-granularity evaluation mechanism, termed evaluation and reflection, was introduced.

**Weaknesses:**

1. Several frameworks have already been proposed where LLMs provide self-feedback and modify their answers based on that feedback [1],[2]. While the implementation details may vary, a detailed explanation is needed on how the fundamental concept of this study differs from previous research.

2. Improved KGQA performance and solving the hallucination problem are distinct issues. There was no specific evaluation of the effectiveness of addressing the hallucination problem.

3. Rather than a two-step process consisting of evaluation and reflection, I think that it could have been more intricately designed as an iterative n-step process. It is unclear why evaluation and reflection must necessarily be limited to two stages, and a fundamental reason for this should be clarified.

[1] Baek, J., Jeong, S., Kang, M., Park, J. C., & Hwang, S. J. (2023). Knowledge-augmented language model verification. arXiv preprint arXiv:2310.12836.
[2] Madaan, A., Tandon, N., Gupta, P., Hallinan, S., Gao, L., Wiegreffe, S., ... & Clark, P. (2024). Self-refine: Iterative refinement with self-feedback. Advances in Neural Information Processing Systems, 36.

**Questions:**

1. What is the fundamental difference between this study and previous research on LLM-based feedback?
2. A detailed explanation is needed on why this feedback process is particularly important in KGQA.
3. Hallucination was pointed out as a problem with LLMs, but has there been no specific evaluation addressing this issue?

---

### Official Review · Reviewer_Jx2Z · 2024-11-01

**Soundness:** 2
**Presentation:** 3
**Contribution:** 1
**Rating:** 3
**Confidence:** 3

**Summary:**

This paper introduces a method called Generation-Evaluation-Reflection (GER) for KBQA tasks, which enhances language model performance in KBQA through three phases.

**Strengths:**

The paper is well-written, with three phases that are easy to understand, helping readers better grasp the main contributions and solution approach.

The authors provide detailed analytical experiments and case studies demonstrating the specific performance and robustness of the Ger method, which enhances the paper's soundness.

**Weaknesses:**

The paper's main weakness is that its contributions don't meet ICLR standards. Specifically, the proposed technique represents an incremental improvement, applying existing technologies including self-evaluation [1] and self-reflection [2]. These have already been proven effective across multiple LLM-related tasks. The authors applied these techniques to the KBQA domain through prompting, which in my view still lacks technical innovation and novelty.

The authors lack comparison and discussion with some existing baselines, such as [3,4,5], which used similar approaches for improvement. However, the authors lack discussion and experiments with these techniques.


---

[1] A Survey on Self-Evolution of Large Language Models

[2]  Self-Reflection in LLM Agents: Effects on Problem-Solving Performance

[3] Generate-on-Graph: Treat LLM as both Agent and KG in Incomplete Knowledge Graph Question Answering

[4] KG-CoT: Chain-of-Thought Prompting of Large Language Models over Knowledge Graphs for Knowledge-Aware Question Answering

[5] Think-on-Graph 2.0: Deep and Faithful Large Language Model Reasoning with Knowledge-guided Retrieval Augmented Generation

**Questions:**

See in Weaknesses.

---

### Official Review · Reviewer_eUcg · 2024-11-02

**Soundness:** 3
**Presentation:** 2
**Contribution:** 2
**Rating:** 3
**Confidence:** 4

**Summary:**

The paper proposes Generation-Evaluation-Reflection (GER), an LLM-enhanced framework specifically designed to address the hallucination problem in KGQA tasks. The key innovation lies in its dual-granularity evaluation mechanism and feedback-based training paradigm. The framework systematically evaluates both the completeness and correctness of candidate answers, using this feedback to guide answer regeneration. Experimental results on WebQSP and CWQ datasets show significant improvements over state-of-the-art methods.

**Strengths:**

The paper addresses a critical challenge in KGQA - the hallucination problem exacerbated by irrelevant information in knowledge graphs.
The dual-granularity evaluation mechanism is well-designed. Question-level evaluation ensures answer completeness; Answer-level evaluation validates individual answer correctness. And the feedback-based training paradigm enables iterative improvement of answers

**Weaknesses:**

1. The paper lacks clear description of instruction data generation process and fine-tuning details.
2. Limited experimental analysis:
- Only basic performance comparison and simple ablation studies
- No analysis of computational costs or efficiency
- Missing important baseline comparisons with recent works like GNN-RAG and GCR
3. The evaluation-reflection mechanism is not fundamentally different from existing self-improvement approaches.

**Questions:**

1. How does the instruction data generation and fine-tuning process differ from RoG? Please provide detailed comparisons.
2. What is the theoretical justification for implementing both question-level and answer-level evaluations? How does the framework handle potential conflicts between these two evaluation levels? For instance, if the answer-level evaluation suggests individual answers are correct but the question-level evaluation indicates incompleteness, what reconciliation mechanism is employed?
3. How does the framework handle cases where the knowledge graph contains conflicting information?
4. Is the framework trained end-to-end or are the modules trained separately? If trained separately, what strategies are employed to ensure coherence between modules? If trained end-to-end, how do you balance the loss functions from different modules?

---

### Official Review · Reviewer_ezUv · 2024-11-05

**Soundness:** 2
**Presentation:** 2
**Contribution:** 2
**Rating:** 3
**Confidence:** 5

**Summary:**

The paper introduces Generation-Evaluation-Reflection (Ger), a framework designed to enhance large language models (LLMs) in Knowledge Graph Question Answering (KGQA). Ger addresses the challenge of hallucinations in LLMs caused by irrelevant knowledge graph data. The framework involves three steps: generation of initial answers, evaluation of answer accuracy and completeness, and reflection to refine answers based on feedback. Experiments on benchmark datasets show that Ger reduces errors and hallucinations, outperforming existing models in KGQA accuracy and reliability, thus setting a new state-of-the-art in this domain.

**Strengths:**

1. The method performs well in two representative benchmarks (WebQSP and CWQ), which validates Ger's effectiveness. Results are clearly articulated, demonstrating how Ger surpasses previous models, particularly the Rog baseline, in key performance metrics like Hits@1 and F1. The experiments also include comprehensive ablation studies to assess the contributions of each module (evaluation and reflection) individually, lending credibility to the robustness of the framework.
2. The presentation of Ger is well-structured, with each component explained in a logical sequence that guides the reader through the framework's mechanism.

**Weaknesses:**

1. Limited Innovation: The method proposed in the paper aligns with existing mainstream methods in terms of reflection processes and does not introduce a reflection mechanism specific to the KG scenario. Although there are improvements on two benchmarks, the method fails to provide new insights and considerations for the readers.

2. Incomplete Baseline Models: The paper does not comprehensively compare methods combining LLM and KG, such as representative works like ChatKBQA[1] and StructGPT[2].

3. Limited Generalizability: The paper does not conduct experiments on a broader range of datasets, making it difficult to demonstrate the method's generalizability, especially in scenarios where large models have been fine-tuned. For example, in different KGs (wikidata) and different types of questions (GrailQA[3]).

[1] ChatKBQA: A Generate-then-Retrieve Framework for Knowledge Base Question Answering with Fine-tuned Large Language Models
[2] StructGPT: A General Framework for Large Language Model to Reason over Structured Data
[3] Beyond I.I.D.: Three Levels of Generalization for Question Answering on Knowledge Bases

**Questions:**

See the Weaknesses section.

---

### Official Review · Reviewer_azBR · 2024-11-08

**Soundness:** 2
**Presentation:** 2
**Contribution:** 2
**Rating:** 5
**Confidence:** 3

**Summary:**

The paper proposes a GER (Generation-Evaluation-Reflection) framework for knowledge graph question answering. The authors’ main argument is that the complex entities and relationships in KGs introduce irrelevant information, which further exacerbates the hallucination problem of LLMs. The paper’s solution is to leverage three modules including generation, evaluation, and reflection. The evaluation module provides feedback to the answers and reasoning paths from the generation module, which are further referred by the reflection module to regenerate answers.

**Strengths:**

(1) The paper identifies the hallucination problem and provides a systematical framework to alleviate it. This can inspire the community to conduct more in-depth research into the problem.

(2) The paper achieves leading performance on the datasets the evaluated on. Ablation studies reveal the effectiveness of different modules in their design.

**Weaknesses:**

(1) As the author noticed in the conclusion section, the method introduces additional computational costs. Importantly, it is widely recognized that scaling inference computes improves the models’ performance in different ways. For example, https://arxiv.org/abs/2408.03314 concludes that sampling more answers for self-verifying is a straightforward way to scale up. Thus, comparing to related methods under similar computational burden may be necessary to support the advantage of the proposed model.

(2) Generalization capability: the paper observes that fine-tuning on the WebQSP and CWQ training set yields improvement. I wonder if this result generalizes to other datasets. If the evaluation and reflection module achieve the general ability to judge the path correctness and provide feedback, there may be improvements on other few-shot datasets without a training set.

(3) The presentation could be improved: in Section 4, training and inference details are missing for necessary reproduction for the proposed method.

**Questions:**

Please refer to weaknesses.

Typos example:
Line 317: Table Table 1 => Table 1

---

### Meta-Review · Area_Chair_UdLE · 2024-12-19

**Metareview:**

There are many concerns about the proposed method, including additional computational costs, limited generalizability across datasets, incomplete baseline comparisons, lack of innovation , insufficient experimental analysis, and missing details. Unfortunately, the authors did not participate in the rebuttal process or address any questions from the reviewers, leading me to recommend rejecting the paper.

**Additional Comments On Reviewer Discussion:**

NIL

---

### Decision · Program_Chairs · 2025-01-22

Reject